# Evolution in Endoscopic Sinus Surgery: The Adjuvant Role of Reboot Surgery in Patients with Uncontrolled Nasal Symptoms of Eosinophilic Granulomatosis with Polyangiitis

**DOI:** 10.3390/jpm13040647

**Published:** 2023-04-09

**Authors:** Francesca Pirola, Gianmarco Giunta, Giovanna Muci, Francesco Giombi, Emanuele Nappi, Fabio Grizzi, Enrico Heffler, Giovanni Paoletti, Giorgio Walter Canonica, Giuseppe Mercante, Giuseppe Spriano, Jan Walter Schroeder, Luca Malvezzi

**Affiliations:** 1Otorhinolaryngology Head & Neck Surgery Unit, IRCCS Humanitas Research Hospital, Via Manzoni 56, Rozzano, 20089 Milan, Italy; francesca.pirola@humanitas.it (F.P.);; 2Department of Biomedical Sciences, Humanitas University, Via Rita Levi Montalcini 4, Pieve Emanuele, 20090 Milan, Italy; gianmarco.giunta.gg@gmail.com (G.G.);; 3Personalized Medicine, Asthma and Allergy Unit, IRCCS Humanitas Research Hospital, Via Manzoni 56, Rozzano, 20089 Milan, Italy; 4Department of Immunology and Inflammation, IRCCS Humanitas Research Hospital, Via Manzoni 56, Rozzano, 20089 Milan, Italy; 5Department of Allergology and Immunology, Ospedale Metropolitano Niguarda Ca’ Granda, Piazza Ospedale Maggiore 3, 20162 Milan, Italy; 6Otorhinolaryngology Head & Neck Surgery Unit, Casa di Cura Humanitas San Pio X, Via Francesco Nava 31, 20159 Milan, Italy

**Keywords:** eosinophilic granulomatosis with polyangiitis, reboot surgery, rhinology, eosinophils, monoclonal antibody, sinus surgery

## Abstract

Purpose: In the era of precision medicine, target-therapy with monoclonal antibodies (mAb) has enabled new treatment options in patients affected by eosinophilic granulomatosis with polyangiitis (EGPA). Nevertheless, sometimes unsatisfactory results at a nasal level may be observed. The aim of this study is to describe reboot surgery as a potential adjuvant strategy in multi-operated, yet uncontrolled EGPA patients treated with Mepolizumab. Methods: We performed reboot surgery on EGPA patients with refractory CRSwNP. We obtained clinical data, nasal endoscopy, nasal biopsy, and symptom severity scores two months before surgery and 12 months after it. Computed tomography (CT) prior to surgery was also obtained. Results: Two patients were included in the study. Baseline sinonasal disease was severe. Systemic EGPA manifestations were under control, and the patients received previous mepolizumab treatment and previous surgery with no permanent benefits on sinonasal symptoms. Twelve months after surgery, nasal symptoms were markedly improved; endoscopy showed an absence of nasal polyps and there were fewer eosinophils at histology. Conclusions: We presented the first experience of two EGPA patients with refractory CRSwNP who underwent non-mucosa sparing (reboot) sinus surgery; our results support the possible adjuvant role of reboot surgery in this particular subset of patients.

## 1. Introduction

Eosinophilic granulomatosis with polyangiitis (EGPA) is an antineutrophil cytoplasmic antibody (ANCA), associated with systemic small-vessel vasculitis characterized by eosinophil-rich necrotizing granulomatous inflammation and hypereosinophilia [1]. EGPA is a systemic disease which may affect virtually any body site, including the lungs, upper and lower airways, nervous system, skin and kidneys as well as gastrointestinal and cardiovascular systems. The respiratory tract (including the upper airways) is one of the most frequently involved in localization and nearly all EGPA patients (>90%) have comorbid asthma, which is typically severe and usually precedes EGPA diagnosis by approximately a decade [2,3,4]. Upper airway involvement might manifest as nasal congestion, discharge, bleeding, and facial pain and/or pressure. A history of recurrent sinusitis is frequently reported, as about 60% of EGPA patients are affected by chronic rhinosinusitis with nasal polyps (CRSwNP) [5]. As for asthma, the onset of sinonasal disease often precedes the diagnosis of EGPA; in this time frame, patients that will eventually be diagnosed with EGPA and may be referred for surgical CRSwNP treatment, but a very high tendency for early recurrence following endoscopic sinus surgery (ESS) has been reported in this setting [5]. Accordingly, a study by Cottin et al. estimated that the local disease relapse rate after surgery was 28% in EGPA patients who underwent standard surgical treatment for CRSwNP [6].

The standard systemic treatment for EGPA consists of oral corticosteroids (OCS) and immunosuppressants (e.g., cyclophosphamide, rituximab, azathioprine, mycophenolate mofetil, or methotrexate) [7]. More recently, a multicenter phase 3, double-blind, placebo-controlled trial (MIRRA trial) demonstrated that Mepolizumab, an anti-IL-5 monoclonal antibody, is effective for the treatment of EGPA at a dose of 300 mg every 4 weeks [8]. The most recent American College of Rheumatology (ACR)—Vasculitis Foundation (VF) guidelines recommend add-on therapy with Mepolizumab for patients with refractory non-severe disease manifestations (such as sinonasal symptoms) [7]. Additional measures to treat sinonasal involvement include topical therapies such as nasal rinses with normal saline and intranasal corticosteroids [9]. The approaches described above are highly effective in the majority of EGPA cases, but some patients experience refractory sinonasal disease despite optimal local and systemic therapy. Although CRSwNP-related symptoms might not be life-threatening, as with other EGPA manifestations, they still represent a significant burden for patients given their great impact on quality of life [10].

The purpose of this study is to evaluate if non-mucosa sparing (reboot) ESS can have an adjuvant role in EGPA patients with refractory sinonasal disease. Reboot surgery, as described by Alsharif et al. [11], is classified as *full* or *partial*, according to whether a Draf III frontal senotomy is performed. The removal of diseased mucosa, in a particular subset of patients affected by recalcitrant drug-resisting type 2 CRSwNP, has proven to be more effective in obtaining disease control as compared to the standard mucosa-sparing technique, improving both recurrence free survival, quality of life outcomes, and the need for OCS treatment [12,13]. This approach aims at restoring the physiologic muco-ciliary mucosal pattern, by removing the panel of inflamed nasal mucosa up to the periosteum of the ethmoidal, sphenoid, and maxillary sinuses [11,12].

In the era of precision medicine and personalized treatments, we advocate that it is useful to investigate whether selected patients’ subsets (e.g., those with comorbid EGPA and CRSwNP) could benefit from this innovative surgical strategy.

## 2. Patients and Methods

The study was approved by our Institutional Review Board (protocol number: ICH/320/21), and it followed the indications of the Declaration of Helsinki and its later amendments. All participants have signed informed consent for the collection and use of their clinical data. Two female patients, 56 and 59 years old, were referred to our Otorhinolaryngology Department. They received a diagnosis of EGPA, respectively, 6 and 3 years prior to referral and had been under treatment with Mepolizumab for 12 months since diagnosis, at dosage of 300 mg every 4 weeks. Both patients’ treatment was discontinued due to symptom control failure and lack of patient compliance. At referral, medical history was collected, the 22-item Sinonasal Outcome Test (SNOT-22) and a Visual Analogue Scale (VAS) for all nasal symptoms were determined. The Nasal Polyp Score (NPS) was assessed through a nasal endoscopy and the Lund-MacKay score (LM) with a maxillo-facial computed tomography (CT) scan. In addition, an ethmoidal mucosa biopsy was performed pre-operatively under local anesthesia (Table 1). Both patients were locally unresponsive to maximal medical therapy and were eventually considered eligible for intervention; therefore, reboot surgery was performed 2 months after referral. Post-operative follow-up was conducted using the same surgical equip. At examination after 12 months, VAS and SNOT-22 scores were again administered and compared to pre-operative data. We did not consider performing a follow-up CT scan due to the lack of subsequent clinical indications. A new biopsy of ethmoidal mucosal tissue was performed, and histologic specimens were compared at high-field magnification. Post-operative medical therapy consisted of daily rinses with saline solution multiple times a day, as well as steroid nebulization twice a day (budesonide 0.5 mg/mL). In neither patient was adjuvant administration of OCS required for the local control of symptoms. In consideration of the sample size, statistical analysis for significance was not assessed.

## 3. Results

Both patients had previously undergone multiple endoscopic sinus surgeries (patient #1 = 7; patient #2 = 4), with the early recurrence of symptoms soon after the last procedure (at 6 and 8 months, respectively). At the time of referral, peripheral eosinophilia was within normal ranges (patient #1 = 100 × 10^3^/mL; patient #2 = 180 × 10^3^/mL) and previous EGPA manifestations (uncontrolled bronchial asthma, lower limbs polyneuropathy, chronic vasculopathy, lung ground-glass opacities) were absent, demonstrating a stable control of systemic disease; even so, nasal symptoms were still severely affecting both patients (Table 1), hence they improperly self-administered OCS without consistent results.

The CT scan (Figure 1) and endoscopic findings (Figure 2) demonstrated massive involvement and nasal obstruction due to the presence of polyps. Pre-operative biopsy revealed a markedly high density of eosinophils in both patients, greater than 10 per high power field (HPF)—patient #1: 10/HPF; patient #2: 12/HPF—despite ongoing treatment with topical steroids, frequent OCS use, and previous mepolizumab therapy. At examination after 12 months, the nasal specimen showed markedly reduced signs of infiltration by eosinophils (≤1/HPF; Figure 3). Likewise, nasal symptoms and signs were still well controlled (Table 1), and no OCS assumption was deemed by patients to achieve clinical control of disease. Finally, at nasal endoscopy, regenerated healthy mucosa were observed overlying the sinuses, with no sign of polyp recurrency (Figure 2).

## 4. Discussion

To date, no standardized guidelines for the management of CRSwNP in EGPA patients have been approved. Although conventional therapies are often capable of controlling systemic EGPA manifestations with excellent outcomes, nasal involvement might have an independent behavior and underlying mechanism, explaining the persistence of sinonasal disease notwithstanding optimal therapy in some patients, even when other systemic disease manifestations are in remission. In a 2018 study by Seccia et al. [10], involving 39 EGPA patients with adequate systemic control with medical therapies according to a rheumatological evaluation, it emerged that nearly all patients (97.4%) had active and persistent nasal symptoms, 71.8% of whom had chronic rhinosinusitis, frequently related to the presence of nasal polyps (46.2%), with a significant impact on quality of life. Moreover, the literature suggests the poor outcomes of conventional surgical approaches for CRSwNP treatment in EGPA patients, as early local disease recurrence is frequent with a significant impact on quality of life [5,6,10].

Here, we described two cases of EGPA that failed to obtain sinonasal disease control, despite numerous surgical attempts and even when medical therapy was effective in obtaining the remission of other disease-specific manifestations. This is likely due to the persistence of inflammatory infiltrates within the surgically spared mucosa; it is then reasonable to wonder whether these patients would benefit from extended nasal surgery as adjuvant therapy. Reboot surgery is an extended type of ESS that, besides re-establishing the ventilation and drainage of paranasal sinuses and scar tissue clearance, consists of removing of all inflamed mucosa down to the periosteum of the ethmoidal, sphenoid, and maxillary sinuses. Reboot procedures have already been proven to drastically reduce the recurrence of nasal polyps compared to mucosal-sparing ESS in a non-EGPA multi-operated population with recalcitrant CRSwNP [11,12]. Its added value consists specifically of the mechanical removal of chronically accumulated eosinophilic infiltrates, together with the microbiota and intramucosal germs. In EGPA patients, this may allow a better and more long-lasting local control of nasal manifestations in cases in which standard medical therapy is demonstrated to be unsatisfactory.

Our patients reported major and stable improvements in sinonasal symptoms one year following reboot surgery, as documented by a reduction in SNOT-22 and VAS scores (Table 1). In parallel, during the 12-month nasal endoscopy, regenerated epithelium was observed in both patients, covering nasal fossae whin no sign of polyp recurrency (NPS = 0; Figure 2). Moreover, at that time, histological samples showed a marked reduction of eosinophilic infiltrates (Figure 3).

Given the rationale behind sinuses demucosization and the optimal results these two patients experienced (that thus far persisted for twice the time of that which occurred after previous surgeries), the query of whether reboot surgery has an adjuvant role in the context of patients with refractory sinonasal manifestations should be raised. To the best of our knowledge, this is the first study concerning the role of reboot surgery in EGPA patients with CRSwNP.

Notably, our two cases frequently resorted to the use of OCS for otherwise uncontrollable sinonasal disease manifestations before the extended ESS. Indeed, most EGPA patients use a very high cumulative dose of systemic corticosteroids throughout the disease course, implying a significant burden of steroid-related side effects in this population. As a matter of fact, steroid-sparing strategies are also currently a focus of clinical investigations in patients with ANCA-associated vasculitis [8,14,15]. In cases of unsuccessful standard ESS, repeated courses of systemic steroids are not justified by the persistence of sinonasal symptoms alone. We speculate that an improved surgical approach not only leads to the longstanding resolution of refractory sinonasal disease, with relevant implications for patients’ quality of life, but it might also reduce the need to resort to OCS in this setting.

Unfortunately, we were able to include only two patients in the study, so our results should be interpreted in the context of a clinical experience and are not suitable for drawing definitive conclusions. Moreover, patients were referred to our unit only for ENT surgical evaluation and follow-up, which by protocol lasted 1 year. Further investigations with larger patient cohorts, a control group receiving standard sinonasal surgery, and a longer follow-up are needed to define the potentially crucial role of reboot surgery in aiding the long-term control of sinonasal symptoms in locally uncontrolled EGPA patients.

## 5. Conclusions

In the era of personalized medicine, targeted therapy will become the treatment of choice in systemic immune-mediated chronic conditions. However, in the case of a lack of complete response, clinicians should consider alternative therapeutic strategies. In EGPA patients, nasal involvement can behave independently of systemic disease control and might not respond to standard optimal therapies. We presented the first, albeit preliminary, experience of two EGPA patients treated with Mepolizumab, but still with suboptimal results at a nasal level, who underwent non-mucosa sparing (reboot) sinus surgery; our results support the possible adjuvant role of this approach in a particular subset of patients, with potential benefits in quality of life and ending the need to resort to OCS to treat refractory sinonasal disease manifestations. ENT surgeons need to re-elaborate their thinking about extended ESS (reboot), deviating from standard mucosa-sparing techniques that are likely to be ineffective, but further research is needed to draw definitive conclusions.

## Figures and Tables

**Figure 1 jpm-13-00647-f001:**
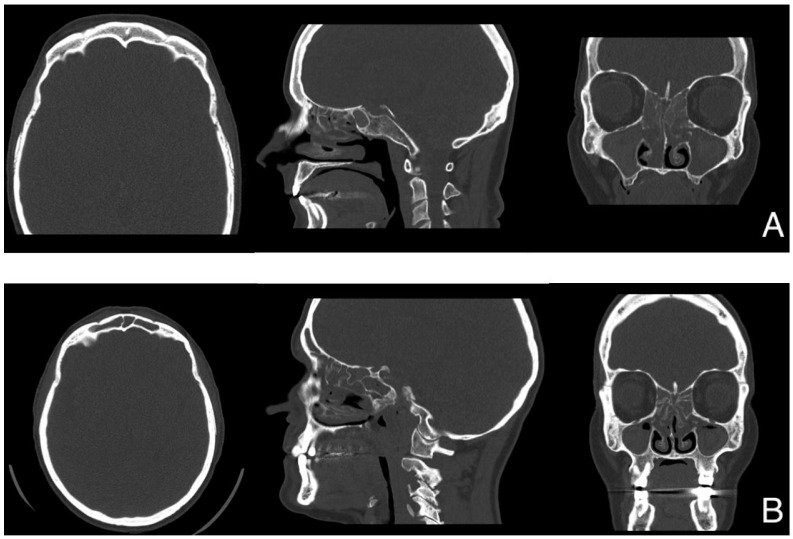
CT scans prior to surgery. (**A**) = patient #1 (frontal sinuses aplasia, LM = 18/20); (**B**) = patient #2 (LM = 22/24).

**Figure 2 jpm-13-00647-f002:**
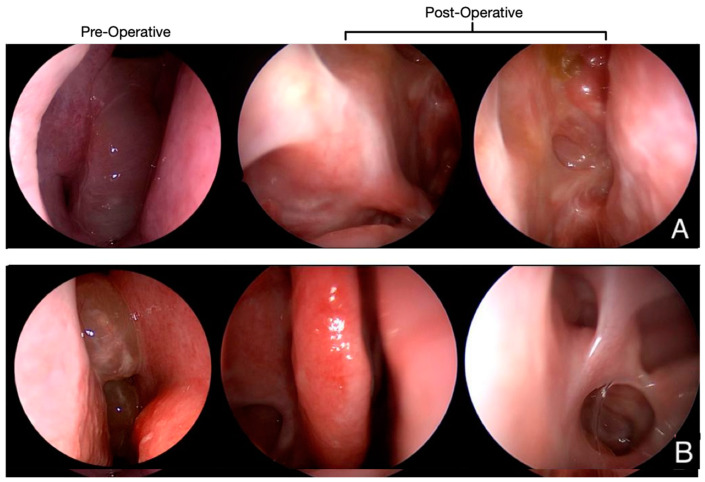
Pre- and post-operative nasal endoscopy (right nasal fossa). (**A**) = patient #1. Pre-operatively, obstructive polypoid formation is observed (NPS = 4). At examination after 12 months, nasal epithelial mucosa had been restored and no recurrence of nasal polyps is observed (NPS = 0). Middle turbinate was sacrificed. Middle antrostomy as well as ethmoidectomy are still open. Surgical scars obliterate sphenoidotomy, with no purulent discharge; (**B**) = patient #2. Preoperatively, sub-obstructive polypoid formations were shown (NPS = 3). At 12-month follow-up, healthy epithelium covers the whole surface of nasal fossa (NPS = 0). Surgical sinusotomies are wide open and correctly draining.

**Figure 3 jpm-13-00647-f003:**
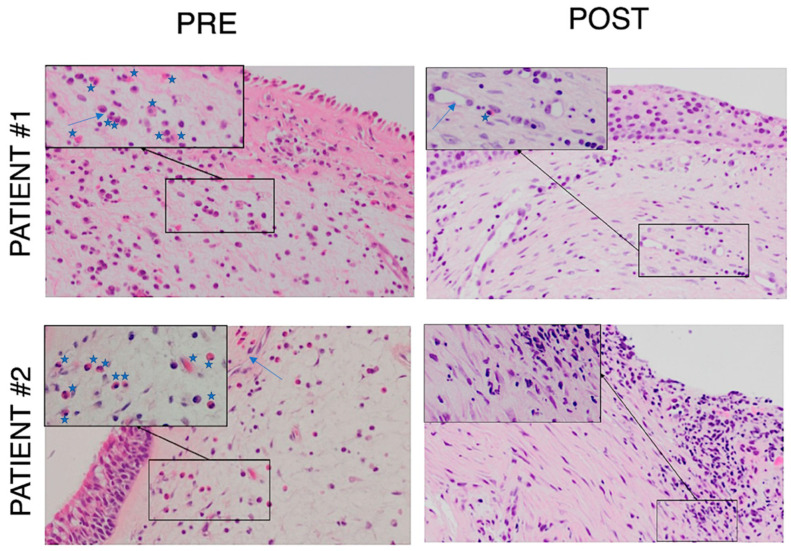
PRE: Histological samples 2 months prior to surgery (PRE), showing typical EGPA peri-vascular eosinophilic pattern (patient #1: 10/HPF; patient #2: 12/HPF). POST: Histological samples 12 months after reboot surgery depict markedly reduced eosinophilic infiltration (≤1/HPF); objective magnification at 20× (at 40× in the boxes); star: eosinophil, arrow: vessel.

**Table 1 jpm-13-00647-t001:** Outcomes 2 months prior (PRE) and 12 months after reboot surgery (POST).

	Patient #1		Patient #2	
	PRE	POST	PRE	POST
Lund-Mackay score	18/20 *	n.a.	22/24	n.a.
Nasal endoscopy	NPS 3 (L)	NPS 0 (L)	NPS 3 (L)	NPS 0 (L)
	NPS 4 (R)	NPS 0 (R)	NPS 3 (R)	NPS 0 (R)
SNOT-22	56	11	104	21
VAS (out of 10)				
Nasal obstruction	7	0	9	0
Hyposmia	10	0	9	3
Dysgeusia	10	0	7	3
Headache or facial pain/pressure	0	0	9	0
Anterior rhinorrhea	7	0	5	1
Posterior rhinorrhea	6	1	5	2

* LM score denominator was downgraded to 20 due to frontal sinuses aplasia.

## Data Availability

The data that support the findings of this study are available from the corresponding author [Giovanna Muci], upon reasonable request.

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
