# Peer review of "Evolution in Endoscopic Sinus Surgery: The Adjuvant Role of Reboot Surgery in Patients with Uncontrolled Nasal Symptoms of Eosinophilic Granulomatosis with Polyangiitis"

_jpm, 2023, doi:10.3390/jpm13040647_

Round 1

Reviewer 1 Report

I appreciate the opportunity to review the manuscript for publication in MDPI JPM.

I feel that this brief paper is well-organized and the present configuration may be worthy of publication in the JPM based on the editor's discretion. Although the summarized results draw attention with comprehensive results as well as recent reference surveys,

further research is required for validation study to substantiate the results.

I have a few comments.

Figure 1: Please describe the degrees of eosinophilic infiltration in all presented samples.

Further, time-course changes in peripheral Eos should be displayed.

L106: Follow-up was conducted by the same surgical equip.

Please explain details in medical therapy after surgery, especially the uses of OCS.

L129: The CT scan and endoscopic findings demonstrated massive involvement and nasal obstruction due to the presence of polyps.

The authors should present these actual Figures for better understanding.

Reviewer 2 Report

The paper, titled as “Evolution in endoscopic sinus surgery: the adjuvant role of Reboot surgery in patients with uncontrolled nasal symptoms of Eosinophilic Granulomatosis with Polyangiitis”, by Francesca Pirola et al, to explore a new therapy methods with EGPA. It suggested that reboot surgery may help in aiding control of sinonasal symptoms. This research is interesting. However, there are something that needed to clarify before drawing some conclusions.

1. This study involves only two cases and lacks persuasiveness.

2. The follow-up time is only one year, and the possibility of recurrence cannot be observed.

3. Lack of endoscopic image data.

Round 2

Reviewer 2 Report

The manuscript has been revised to be suitable for publication.